# Coronavirus disease 2019 (COVID-19) vaccine acceptability in Ghana: An urban-based population study

**Hannah Benedicta Taylor-Abdulai**[1☉], **Edem Kojo Dzantor**[2,3☉], **Nathan Kumasenu Mensah**[4*☉],
**Mubarick Nungbaso Asumah**[5], **Stephen Ocansey**[1], **Samuel Kofi Arhin**[1], **Precious Barnes**[1],
**Victor Obiri Opoku**[1], **Zakariah Jirimah Mankir**[1], **Sylvester Ackah Famieh**[1],
**Collins Paa Kwesi Botchey**[1]

**1** Department of Physician Assistant Studies, School of Allied Health Sciences, University of Cape
Coast, Cape Coast, Ghana, **2** Department of Epidemiology and Biostatistics, Fred N. Binka School of
Public Health, University of Health and Allied Sciences, Hohoe Campus, Hohoe, Ghana, **3** Research and
Innovation Unit, College of Nursing and Midwifery, Nalerigu, Ghana, **4** Department of Health Information
Management, School of Allied Health Sciences, University of Cape Coast, Cape Coast, Ghana, **5** Nurses
and Midwifery Training College, Ministry of Health, Tamale, Ghana

☉ These authors contributed equally to this work
* nathan.mensah@ucc.edu.gh

journal.pone.0319798

and Health Sciences University: Ras Al
Khaimah Medical and Health Sciences
University, UNITED ARAB EMIRATES

**Peer Review History:** PLOS recognizes the
benefits of transparency in the peer review
process; therefore, we enable the publication
of all of the content of peer review and
author responses alongside final, published
articles. The editorial history of this article is
available here: https://doi.org/10.1371/journal.
pone.0319798

## Abstract

### Introduction

Coronavirus disease 2019 (COVID-19) vaccine hesitancy is a complex health challenge
characterized by a delay in the acceptance or refusal of the vaccination with context-
specific determinants. Our study, therefore, assessed the COVID-19 vaccine acceptance
among urban dwellers in the Central Region, of Ghana.

### Methods

A cross sectional study was conducted between September and November, 2022 using a
multi-stage cluster sampling procedure among 377 participants. A modified World Health
Organization pretested paper-based questionnaire was administered to study participants.
The data was analyzed using Statistical Package for Social Sciences (SPSS) version 26.
Descriptive and inferential statistics were carried out and results were summarized into
frequencies, percentages, tables, and charts for clarity. A conventional p-value < 0.05 was
considered statistically significant.

### Results

The study revealed that COVID-19 vaccine acceptance was 20.0% (76/377) and vaccine
hesitancy was 80.0% (301/377). Out of the 377 participants, their socio-demographic
characteristics showed that the majority were below 25 years 53.8% (203/377), [vaccine
acceptance; 36.84% (28/76) vs vaccine hesitancy; 58.14% (175/301)], and females 50.1%
(189/377), [vaccine acceptance; 56.58% (43/76) vs vaccine hesitancy; 48.50% (146/301)].
Common reasons for COVID-19 vaccine hesitancy included mistrust of the source of

**Data availability statement:** Data can be shared publicly and has been made available.

**Funding:** The author(s) received no specific funding for this work.

**Competing interests:** The authors have declared that no competing interests exist.

the vaccine, personal belief and experience, mistrust of the drug development process, mistrust in the health system, and mistrust of the pharmaceutical company. Age above 25 years, female, educational levels, senior high school and above, being employed, and hearing of new vaccine had a significance influence on COVID-19 vaccine acceptance.

## Conclusion

COVID-19 vaccine acceptance was low with high vaccine hesitancy among participants. The study's findings highlights the importance of addressing vaccine hesitancy through building trust in the vaccine development processes, including the provision of accurate information about the vaccine safety and efficacy. Resolving concerns related to the source of the vaccine and the overall healthcare system are important to address vaccine hesitancy. Policy makers could adopt tailored interventions targeting specific demographic groups, such as the younger population and females to increase vaccine acceptance. Ghana's public health authorities could adopt the findings to re-strategize its urban COVID-19 vaccine campaigns to address misconceptions and misinformation to increase vaccine acceptance.

## Introduction

The world is currently fighting the "double burden" of Coronavirus disease (COVID-19); that is the spread of COVID-19 and COVID-19 vaccine hesitancy [1–3]. According to the World Health Organization (WHO) dashboard on COVID-19, the global estimates of confirmed cases of COVID-19 were 754,018,841 and 6,817,478 deaths as of 3 February, 2023 [4]. Similarly, the WHO indicated that as of 31 January 2023, 13,168,935,724 COVID-19 vaccine doses have been administered globally [4]. COVID-19 vaccine hesitancy has become a major challenge globally [5–8] and perhaps the next "plague" the world must come together to address especially in the African region. The WHO describes vaccine hesitancy as the 7th among the top 10 threats to global public health and a major contributor to the spread of vaccine preventable diseases in Africa [9]. The COVID-19 vaccine acceptance rate varies remarkably across and within many countries globally. Njoga and colleagues in a systematic review of the COVID-19 vaccine acceptance rate from 2021 to 2022 showed that the overall vaccine acceptance rate varied between 21.0% to 97.9% (59.8 ± 3.8%) in 2021 and 8.2% to 92.0% (58.0 ± 2.4%) in 2022 [9]. The study equally showed that the Southern and Eastern African regions recorded the top two vaccine acceptance rates. They indicated that as of 25 October 2022, Africa had recorded 24% fully COVID-19 vaccinated persons paralleled to the Australian continent (84%), upper-middle-income countries (79%), and globally (63%) [9].

COVID-19 vaccine hesitancy is a complex health challenge characterized by a delay in the acceptance, refusal of vaccination, issues of vaccine confidence, adverse events, complacency, convenience, communications, and the context in which it is administered [9–11]. Identifying the major context-specific determinants that drive COVID-19 vaccine acceptance becomes an important approach to addressing vaccine hesitancy and increasing vaccine acceptance rates. Wiysonge and colleagues asserted, examining the context-specific determinants of COVID-19 vaccine acceptance is important for tailor-made public health policies and interventions that effectively meet and address challenges of vaccine hesitancy peculiar to the setting [12]. As of 30th January, 2023, the Ghana Health Service COVID-19 dashboard reported that 23,226,767 doses of vaccines have been administered with 9,909,398 fully vaccinated individuals

representing 54.3% of a target population of 18 million [13]. Varied determinants of COVID-19 vaccine acceptance have been assessed in Ghana [1,5,14]. Several of the studies are limited geographically and among sections of the Ghanaian population including health professionals [1,14]. Other population based studies have described the willingness to take the COVID-19 vaccine [15,16]. Studies have reported disparities in COVID-19 vaccine acceptance among urban and rural populations, with most of the studies reporting lower vaccination rates in rural population in West African countries [17,18]. However, [19] reported significant COVID-19 vaccine hesitance among urban participants compared to rural participants. The dynamics in COVID-19 vaccine acceptance among urban and rural populations may differ, requiring different remedial approaches to address issues of vaccine hesitancy and increase in vaccine acceptance.

Several studies have utilized the Health Belief Model (HBM) to understand COVID-19 vaccine acceptability. For instance, a study by Wong et al. (2021) applied the HBM to access vaccine acceptance in Malaysia, finding that perceived susceptibility, severity, belief, and barriers significantly influenced individuals' intentions to get vaccinated [20]. Similarly, a study conducted by Shmueli (2021) in Israel used the HBM to predict the intention to receive the COVID-19 vaccine, highlighting the importance of perceived benefits and cues to action [21]. In Egypt, a cross-sectional study using the HBM identified that higher perceived severity and susceptibility were associated with increased vaccine acceptance [22]. These studies underscore the value of the HBM in identifying the psychological and social factors that can be targeted to improve vaccine uptake.

The current determinants of COVID-19 vaccine acceptance in the Central Region of Ghana are yet to be fully explored. The Central Region is a cosmopolitan region with a population of 2,859,821, with the majority being urban dwellers and COVID-19 vaccine acceptance rate of 50.4% [13]. This suggests that about 49.6% of the population in the region are yet to be vaccinated. This study therefore, examined the COVID-19 vaccine acceptance and its associated factors in an urban setting with diverse population in the Central Region of Ghana. Unlike other studies that examined willingness and attitudes towards COVID-19 vaccine uptake before the rollout of the vaccine exercise in Ghana, the current study used real-world data from the COVID-19 vaccine exercise initiated more than a year ago. The findings of this study are crucial to inform context-specific COVID-19 vaccine acceptance initiatives to address the challenges and promote vaccine activities.

## Materials and methods

### Ethical considerations

Ethical approval was granted for the study protocol by the University of Cape Coast Institutional Review Board (Approval number – UCCIRB/EXT/2021/40). As part of the data collection, each participant provided written consent and signed an informed consent form before participation in the study. The informed consent was obtained without coercion, undue influence or misrepresentation of potential benefits and risk associated with participation in the study. Participants were assured that their participation was entirely voluntary. No personal identifiers, such as names or designations of the participants was collected to ensure anonymity of the participants. Hence, there was no information that could be used to identify individual participants.

### Study design

A descriptive cross-sectional, paper-based quantitative survey was conducted within 2 months period from September to November 2022.

## Study site

The study was conducted in the Cape Coast Metropolitan area. The metropolitan area is the only metropolis out of twenty three (23) districts in the Central region. It shares boundaries with the Gulf of Guinea to the south, Komenda Edina Eguafo Abrem Municipal District to the west, Abura Asebu Kwamankese District to the east and the Twifu Heman Lower Denkyira District to the north. The area occupies approximately 122 square kilometres. The total population of Cape Coast Metropolis, according to the 2021 population and housing Census is 189,925. The metropolis is divided into two sub-metropolis: the Cape Coast North and the Cape Coast South sub-metropolis. Each sub-metropolis is further divided into zones. The Cape Coast North sub-metropolis has three (3) zones comprising the Abakam-Ola University, Effutu-Kakomdo-Mempeasem, and Abura-Adisadel-Pedu-Nkafoa. The Cape Coast South sub-metro has four (4) zones comprising Akyim-Amanful-Brofoyedur-Ekon, Gyegyem Instin-Krootown, Anakyin-Baakano-Chapel Square, and Aboom-Esikafoambantsem-Kadadwem [23]. Cape Coast Metropolitan area is the home to many category A senior high schools, the University of Cape Coast, which recently has been rated as the number one university in Ghana according to the Times Higher Education (THE). It is also a major tourist destination in Ghana. This makes Cape Coast Metropolitan area an important place. Arguably, the north is more elitist, hence the selection of the Cape Coast North sub-metropolitan area for the study.

## Sampling techniques and sample size

A multi-stage cluster sampling approach was used. In the first stage, the Cape Coast Metropolis was divided into two sub-metropolitan areas: Cape Coast North and Cape Coast South. The Cape Coast North sub-metropolis was selected for further sampling. Within the Cape Coast North sub-metropolis, there are three zones: Abakam-Ola University, Effutu-Kakomdo-Mempeasem, and Abura-Adisadel-Pedu-Nkafoa. From these zones, Abakam-Ola University and Effutu-Kakomdo-Mempeasem were chosen for the study. In the next stage, enumeration areas within these selected zones were further stratified. Households were then randomly selected from each stratum to ensure representative sampling. The study population consisted of participants aged 18 years and above who were selected from these households.

The computation of the sample size was 384. Based on the population size of 189,925, a minimum sample size of 384 was determined, using the formula,

$$n = \frac{Nx}{x + N - 1}.$$

where $n$ is the sample size,

$N$ - Population size and $\quad x = Z^2 P(1 - P)/W^2$

Where $Z$ is the statistics for a level of 95% confidence interval (CI) = 1.96; $P$ - The assumed proportion of COVID-19 vaccine hesitancy and acceptance is 50% with a margin of error ($W$) of 5%. In the absence of specific data, we assumed the population is equally split between those who might accept and those who might be hesitant about the vaccine. Therefore, 50% served as a neutral baseline to avoid overestimating or underestimating vaccine acceptance or hesitancy.

However, provision was made for non-response, therefore the total data collected was 405 out of which 377 was used for analysis.

## Data collection procedures

A trained research team of ten persons collected the data using paper-based questionnaires. Prior to commencement of the data collection, the aim of the study was explained to the

participants by the research team. A face-to-face administration of questionnaires was adopted. Respondents who could not read English, were assisted by the data enumerators, and those who could read attempted the questions themselves.

## Data collection instruments

We adapted a questionnaire originally developed by the World Health Organization (WHO). The first section of the questionnaire consisted of demographic characteristics of participants such as age, gender, level of education, religion, and occupation. The second section looked at knowledge about the COVID-19 vaccine. The third section asked participants about the various types of COVID-19 vaccines that have been developed so far, the fourth section was on respondents' source of information about the vaccine, and the fifth section was on perceived risk and severity of contracting COVID-19. The final section assessed whether participants' have taken the vaccine and misinformation about the COVID-19 vaccine.

## Data quality control

To ensure data quality, a pretest of the data collection tool was conducted among twenty-five (25) students at the University of Cape Coast. The pretest provided an avenue to assess the content of the items, their comprehensibility, and suitability after which modifications were made based on the feedback from the field. Unique identification numbers was assign to each questionnaire for accountability and for easy recall purposes. An independent team of supervisors double-checked the questionnaires for consistency, accuracy and completeness, after which they were entered into IBM SPSS version 25. Out of the 405 data collected, 377 (93.1% response rate) was used in performing the analysis.

## Data processing and statistical analysis

Analysis of the data was done after checking for accuracy and completeness. A total of 377 participants were included in the data analysis after data cleaning. Data was analyzed using descriptive statistics such as frequency, percentage, and mean. Inferential statistics such as chi-square test was performed to determine the relationship between COVID-19 vaccine uptake or otherwise with participants' Socio-demographic characteristics. Finally, all variables with p values less than 0.25 in the chi-square analysis were included in the logistic regression model, allowing all potentially important predictive factors to be included in the modified model and to control for confounding variables. Therefore, all variables with p values greater than 0.25 were not considered in the logistic regression models. Literature suggests that p values less than 0.3 or p value closer to zero (0) gives a desirable outcome of the goodness of a logistic regression model [24,25]. The benchmark p value of ≤ 0.25 was selected because of its closeness to zero (0) thus enabling an inclusion of all potentially relevant predictive variables in the adjusted model. The odds ratios (OR) were recorded with their corresponding 95% confidence intervals (95% CI). A *p-value* of less than 0.05 was considered statistically significant.

# Results

## Demographic characteristics and distribution of COVID-19 vaccine acceptance

Out of the of 377 participants, their socio-demographic characteristics showed that the majority were below 25 years 53.80% (203/377) [vaccine acceptance; 36.84% (28/76) vs vaccine hesitancy; 58.14% (175/301)], **Females** 50.10% (189/377), [vaccine acceptance; 56.58% (43/76) vs vaccine hesitancy; 48.50% (146/301)], **Christians** 88.60% (334/377) [vaccine acceptance; 88.16% (67/76) vs vaccine hesitancy; 88.70% (267/301)], **had attained tertiary**

**education** 67.10% (253/377) [vaccine acceptance; 85.53% (65/76) vs vaccine hesitancy; 62.46% (188/301)], and **employed** 28.10% (106/377), [vaccine acceptance; 53.95% (41/76) vs vaccine hesitancy; 21.59% (65/301)] (Table 1).

The Chi-square analysis showed significant associations between COVID-19 vaccine acceptance and age (*p = 0.001*), education (*p = 0.001*) and occupation (*p < 0.001*) (Table 1).

## COVID-19 vaccine acceptance among participants

Out of the 377 participants, COVID-19 vaccine acceptance was found to be 20.0% (76) and vaccine hesitancy 80.0% (301).These proportions are depicted in Fig 1

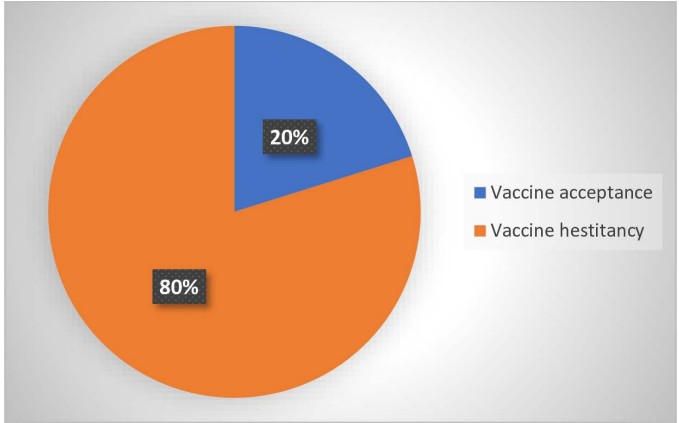

**Fig 1. COVID-19 vaccine acceptance among participants.**

**Table 1. Socio-demographic characteristics and distribution of COVID-19 vaccine acceptance (n = 377).**

| Variables | Category | n = 377 (%) | Vaccine acceptance (76) n(%) | Vaccine hesitancy (301) n(%) | p value |
|---|---|---|---|---|---|
| **Age (years)** | < 25 | 203(53.80) | 28(36.84) | 175(58.14) | 0.001* |
| | ≥25 | 174(46.20) | 48(63.16) | 126(41.86) | |
| **Sex** | Female | 189(50.10) | 43(56.58) | 146(48.50) | 0.208 |
| | Male | 188(49.90) | 33(43.42) | 155(51.50) | |
| **Religion** | Christian | 334(88.60) | 67(88.16) | 267(88.70) | 0.384 |
| | Islam | 37(9.80) | 9(11.84) | 28(9.30) | |
| | Traditionalist | 6(1.60) | 0(0.0) | 6(1.99) | |
| **Education** | No education | 18(4.80) | 3(3.95) | 15(4.98) | 0.001* |
| | Basic education | 36(9.50) | 0(0.0) | 36 (11.96) | |
| | SHS/Voc/Tech | 70(18.60) | 8(10.53) | 62(20.60) | |
| | Tertiary | 253(67.10) | 65(85.53) | 188(62.46) | |
| **Occupation** | Employed | 106(28.10) | 41(53.95) | 65(21.59) | <0.001* |
| | Self employed | 59(15.60) | 7(9.21) | 52(17.28) | |
| | Unemployed | 24(6.40) | 4(5.26) | 20(6.64) | |
| | Student | 188(49.90) | 24(31.58) | 164(54.49) | |
| **Total** | | 377(100.0%) | 76(100.0%) | 301(100.0%) | |

*Significant p-values; Voc = Vocational school, SHS = Senior High School, Technical School.

## Knowledge of COVID-19 and COVID-19 vaccine among participants

Majority of the participants stated that **COVID-19 was real** 89.40% (337/377) [Vaccine acceptance; 96.05% (73/76) vs Vaccine hesitancy; 87.71% (264/301)], **heard of immunity** 86.70% (327/377) [Vaccine acceptance; 93.42% (71/76) vs Vaccine hesitancy; 85.05% (256/301)], **vaccination gives immunity** 36.30% (137/377) [Vaccine acceptance; 48.68%(37/76) vs Vaccine hesitancy; 33.22% (100/301)] and **Vaccination do not prevent other disease** 41.60% (157/377) [Vaccine acceptance; 40.79% (31/76) vs Vaccine hesitancy; 41.86% (126/377)].

Again, most of the participants indicated that **vaccine prevents severity of disease** 58.60% (221/377) [Vaccine acceptance; 67.11%(51/76) vs Vaccine hesitancy; 56.48% (170/301)], **had not experienced COVID-19 symptoms** 62.60% (235/377) [Vaccine acceptance; 64.47% (49/76) vs Vaccine hesitancy; 62.13% (187/301)], **been positive for COVID-19** 6.10% (23/377) [Vaccine acceptance; 10.53% (8/76) vs Vaccine hesitancy; 4.98% (15/301)] and **heard of new vaccine** 60.70% (229/377) [Vaccine acceptance; 84.21% (64/76) vs Vaccine hesitancy; 54.82% (165/301)].

Vaccine acceptance was significant among those with knowledge of vaccination giving immunity (*p < 0.001*) and knowledge of new vaccine (*p < 0.001*) (Table 2).

## Knowledge of common COVID-19 vaccines

The commonly known COVID-19 vaccines included AstraZeneca 60.70% (139/229), Johnson & Johnson 11.80% (27/229), Pfizer 11.40% (26/229), Sputnik V vaccine 6.10% (14/229), Sinovac Biotech 5.70% (13/229) and Modena 4.40% (10/229) as shown in Fig 2.

## Source of information on COVID-19 vaccines

Fig 3 depicts sources of information on COVID-19 vaccine among participants which include Television 83.00% (313/377), Social Media 73.50% (277/377), Radio 70.60% (265/377), Health workers 66.60% (251/377), and Newspaper 65.50% (247/377).

## Perception of participants towards COVID-19

The perception of participants was assessed with more than half 54.10% (204/377), 58.40% (220/377,) indicating that they were at low risk of being and severely infected with COVID-19 respectively. Most of the participants were of the view that they were not all that susceptible of getting COVID-19 36.90% (139/377) and COVID-19 can be severe when contracted 18.30% (69/377) (Table 3).

## Perception towards COVID-19 vaccine acceptance among participants

The common five reasons not to accept the COVID-19 vaccines by the 301 participants included; mistrust of source of vaccine 29.90% (90/301), personal belief and experience 25.90% (78/301), mistrust of drug development process 14.60% (44/301), mistrust in the health system 11.00% (33/301), and mistrust of pharmaceutical company 7.00% (21/301).

More than half of the participants 52.80% (199/377) were not sure that the Ministry of Health (MoH) vaccine roll out is appropriate with 37.10% (140/377) indicating it is appropriate. Sixty-two percent 60.20% (227/377) of the participants were not sure that the COVID-19 vaccine can alter their genetic make-up with 23.60% (89/377) indicating that the vaccine can alter their genetic makeup. Most of the participants were not sure of using COVID-19 vaccines as a means to implant a microchip 49.60% (187/377) with 25.50% (96/377) indicating is a means to implant microchip.

**Table 2. Knowledge of COVID-19 and COVID-19 vaccine among participants (n = 377).**

| Variables | Category | n (%) | Vaccine acceptance (76) n(%) | Vaccine hesitancy (301) n(%) | p value |
|---|---|---|---|---|---|
| **COVID-19 is real** | | | | | 0.074 |
| | Yes | 337(89.40) | 73(96.05) | 264(87.71) | |
| | No | 13(3.40) | 2(2.63) | 11(3.65) | |
| | Not sure | 27(7.20) | 1(1.32) | 26(8.64) | |
| **Heard of Immunity** | | | | | 0.055 |
| | Yes | 327(86.70) | 71(93.42) | 256(85.05) | |
| | No | 50(13.30) | 5(6.58) | 45(14.95) | |
| **Vaccination gives immunity** | | | | | <0.001* |
| | Yes | 137(36.30) | 37(48.68) | 100(33.22) | |
| | No | 119(31.60) | 30(39.47) | 89(29.57) | |
| | Not sure | 121(32.10) | 9(11.84) | 112(37.21) | |
| **Vaccination prevents other disease** | | | | | 0.422 |
| | Yes | 118(31.30) | 28(36.84) | 90(29.90) | |
| | No | 157(41.60) | 31(40.79) | 126(41.86) | |
| | Not sure | 102(27.10) | 17(22.37) | 85(28.24) | |
| **Vaccine prevents severity of disease** | | | | | 0.150 |
| | Yes | 221(58.60) | 51(67.11) | 170(56.48) | |
| | No | 72(19.10) | 14(18.42) | 58(19.27) | |
| | Not sure | 84(22.30) | 11(14.47) | 73(24.25) | |
| **Ever had COVID-19 symptoms** | | | | | 0.691 |
| | Yes | 101(26.80) | 21(27.63) | 80(26.58) | |
| | No | 235(62.60) | 49(64.47) | 187(62.13) | |
| | Not sure | 40(10.60) | 6(7.89) | 34(11.30) | |
| **Been positive for COVID-19** | | | | | 0.071 |
| | Yes | 23(6.10) | 8(10.53) | 15(4.98) | |
| | No | 354(93.90) | 68(89.47) | 286(95.02) | |
| **Heard of new vaccine** | | | | | <0.001* |
| | Yes | 229(60.70) | 64(84.21) | 165(54.82) | |
| | No | 148(39.30) | 12(15.79) | 136(45.18) | |
| **Total** | | 377 (100.0%) | 76(100.0%) | 301(100.0%) | |

*Significant p-values.

About 56.80% (214/377) and 7.20% (27/377) indicated that there were not sure COVID-19 causes infertility and COVID-19 causes infertility respectively. Strategies after COVID-19 vaccine included; use of sanitizer 91.80% (346/377), hand washing 91.50% (345/377), social distance 88.90% (335/377) and stopping shaking hands 85.40% (322/377).

What will make participants to accept the COVID-19 vaccine includes; adequate information about vaccine 36.10% (136/377), positive feedback from vaccinated people 19.90%(75/377), vaccine given for free, 16.70% (63/377), and as a requirement to travel 12.50%(47/377). Most of the participants were willing to recommend COVID-19 vaccines 43.00% (162/377), not willing to recommend 21.50% (81/377) and not sure of recommending the vaccine 35.50% (134/377) (Table 4).

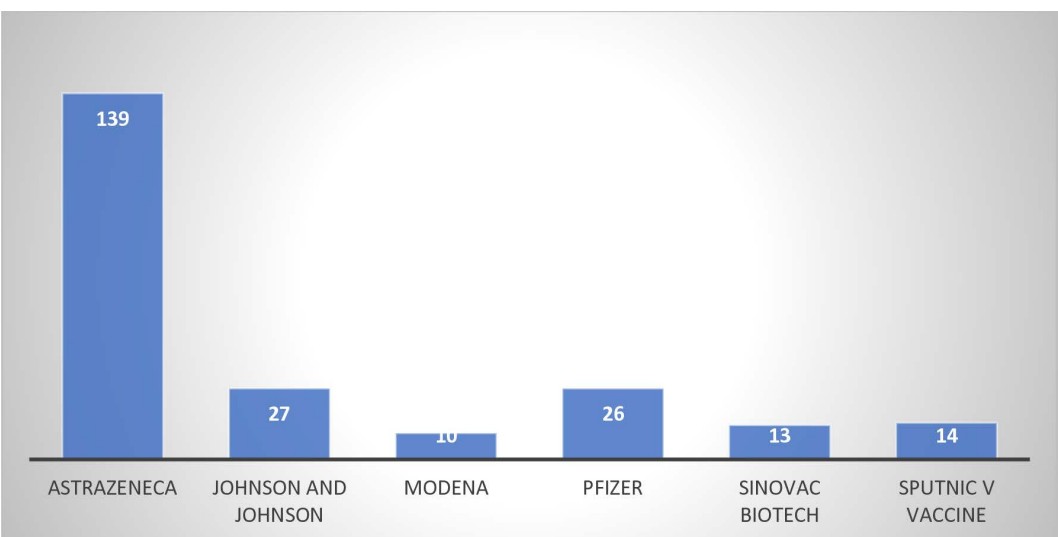

**Fig 2. Knowledge of common vaccines.**

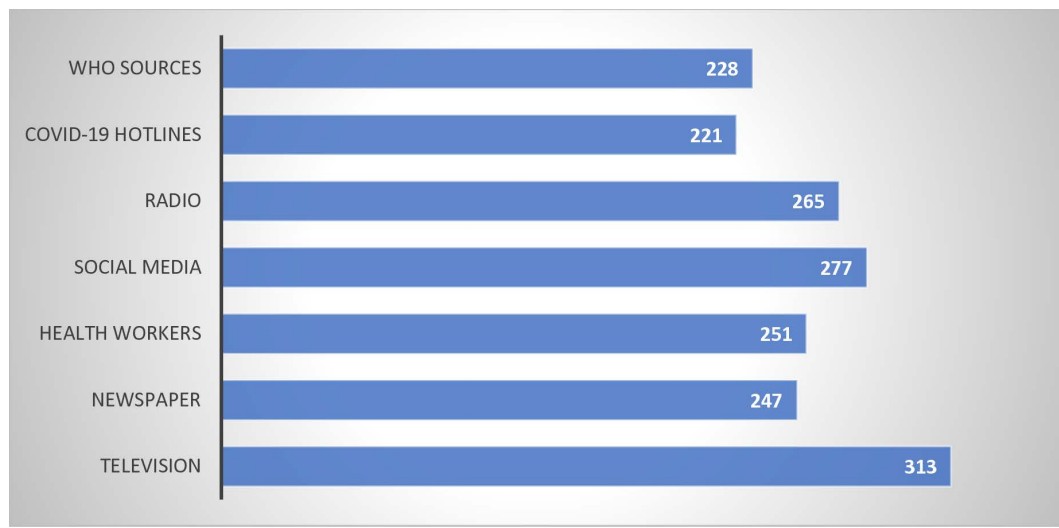

**Fig 3. Source of information on COVID-19 vaccines.**

### Factors associated with COVID-19 vaccine acceptance among participants

The unadjusted odds ratio analysis showed that participants aged ≥ 25 years had increased odds of COVID-19 vaccine acceptance as compared to persons age < 25 years [COR; 2.4, 95%CI (1.4–4.0), p = 0.001]. Participants with educational level from SHS and above showed increased odds of COVID-19 vaccine acceptance as compared to persons with educational level below SHS [COR; 5.0, 95%CI (1.–6.4), p = 0.008]. Participants who are employed are about 4 times more likely to be vaccinated as compared to those who are unemployed [COR; 4.3, 95%CI (2.5–7.2), p < 0.001]. Indication of COVID-19 being real showed increased odds of COVID-19 vaccine acceptance as compared to participants who indicated COVID-19 was not real [COR; 3.5, 95%CI (1.–1.7), p = 0.041]. Participants who indicated that the COVID-19

**Table 3. Perception of participants towards COVID-19 (n = 377).**

| Variable | Categories | Frequency | % |
|---|---|---|---|
| **Risk of being infected** | High unlikely | 92 | 24.40 |
| | Low | 204 | 54.10 |
| | Very high | 81 | 21.50 |
| **Risk of being severely infected** | High unlikely | 100 | 22.50 |
| | Low | 220 | 58.40 |
| | Very high | 57 | 15.10 |
| **Susceptible to COVID-19** | Not at all susceptible | 139 | 36.90 |
| | Not sure | 187 | 49.60 |
| | Very susceptible | 51 | 13.50 |
| **Severity when you contract COVID-19** | Very severe | 60 | 15.90 |
| | Severe | 69 | 18.30 |
| | Not sure | 151 | 40.00 |
| | Not severe | 97 | 25.70 |

vaccination gives immunity are about 1.9 more likely to accept COVID-19 vaccine as compared those who indicated no [COR; 1.9, 95%CI (1.1–3.2), p = 0.013]. Participants who have heard of new vaccine had increased odds of COVID-19 vaccine acceptance as compared those who have not heard of new vaccines [COR; 4.4, 95%CI (2.3–8.5), < 0.001].

The adjusted odds ratio showed that participants aged ≥ 25 years had increased odds of COVID-19 vaccine acceptance as compared to persons age < 25 years [AOR; 2.1, 95%CI (1.1–4.1), p = 0.033]. Females had increased odds of COVID-19 vaccine acceptance as compared to males [AOR; 2.0, 95%CI (1.1–3.6), p = 0.019]. Participants with educational level from SHS and above showed increased odds of COVID-19 vaccine acceptance as compared to persons with educational level below SHS [AOR; 3.3, 95%CI (1.8–13.0), p = 0.044]. Participants who are employed are about 4 times more likely to be vaccinated as compared to those who are unemployed [AOR; 2.5, 95%CI (1.3–4.8), p < 0.005]. Participants who have heard of new vaccine had increased odds of COVID-19 vaccine acceptance as compared to those who have not heard of new vaccines [AOR; 2.9, 95%CI (1.4–6.0), p < 0.004] (Table 5).

## Discussion

The purpose of this study was to assess COVID-19 vaccine acceptance among urban dwellers in Cape Coast, Central Region, Ghana. COVID-19 vaccine acceptance in this study was low, but almost consistent with the 23.0% of vaccine acceptance as reported by Adomako [26], in a cross sectional study among Ghanaian population, that aimed to determine the willingness of the population to accept COVID-19 vaccine. In contrast, Forkuo et al. [27] in their study to assess willingness to accept the COVID-19 vaccine among out-patient department attendants in Bono region in Ghana reported 9.8%, which is less than half what our current study found. These differences are linked to the sample size and population used. This is a common occurence in research; however, the results emphasize the need to increase COVID-19 vaccine uptake by addressing important contextual factors that act as de-motivators. The distribution of COVID-19 vaccine hesitancy among participants showed that the majority were educated and had attained tertiary level of education. The assumption is that all things being equal, educated persons are expected to be more health literate, understand the benefits associated with the vaccine and therefore may take the COVID-19 vaccine as compared to persons with little or no education. This finding emphasizes the necessity to expand COVID-19 vaccine health

**Table 4.  COVID-19 vaccine acceptance among participants.**

| Variable | Categories | Frequency | Percentage |
|---|---|---|---|
| **What informed you not to vaccinate (301)** | | | |
| | Politicians | 7 | 2.30 |
| | Religions leader | 10 | 3.30 |
| | Mistrust of drug development process | 44 | 14.60 |
| | Mistrust health system | 33 | 11.00 |
| | Mistrust origin of vaccine | 18 | 6.00 |
| | Mistrust of source of vaccine | 90 | 29.90 |
| | Personal belief and experience | 78 | 25.90 |
| | Mistrust of pharmaceutical company | 21 | 7.00 |
| **If your concerns are addressed, how soon would you consider taking the vaccine if made available (n = 301)** | | | |
| | Immediately | 56 | 18.60 |
| | After 6 months time | 8 | 2.70 |
| | In about a year | 18 | 6.00 |
| | Not ever | 55 | 18.30 |
| | When am convinced vaccine is safe | 126 | 41.90 |
| | When most people are vaccinated | 38 | 12.60 |
| **MOH vaccine roll out is appropriate** | | | |
| | Yes | 140 | 37.10 |
| | No | 38 | 10.10 |
| | Not sure | 199 | 52.80 |
| **COVID-19 will alter genetic make up** | | | |
| | Yes | 89 | 23.60 |
| | No | 61 | 16.20 |
| | Not sure | 227 | 60.20 |
| **COVID-19 not a means to implant a micro chip** | | | |
| | Yes | 96 | 25.50 |
| | No | 94 | 24.90 |
| | Not sure | 187 | 49.60 |
| **COVID-19 causes infertility** | | | |
| | Yes | 27 | 7.20 |
| | No | 136 | 36.10 |
| | Not sure | 214 | 56.80 |
| **Strategies after COVID-19 vaccine** | | | |
| | Wear face mask | 292 | 77.50 |
| | Social distance | 335 | 88.90 |
| | Hand washing | 345 | 91.50 |
| | Use sanitizer | 346 | 91.80 |
| | Avoid large gatherings | 320 | 84.90 |
| | Stop shaking hands | 322 | 85.40 |
| **What will make you accept COVID-19 vaccine** | | | |
| | Financial incentive | 40 | 10.60 |
| | Monetary rewards to health workers | 16 | 4.20 |
| | Vaccine given for free | 63 | 16.70 |
| | Adequate information about vaccine | 136 | 36.10 |
| | Requirement to travel | 47 | 12.50 |
| | Employment condition | 35 | 9.30 |
| | Positive feedback from vaccinated people | 75 | 19.90 |

*(Continued)*

**Table 4.** (Continued)

| Variable | Categories | Frequency | Percentage |
|---|---|---|---|
| **Would you recommend COVID-19 vaccine to your friends** | | | |
| | Yes | 162 | 43.00 |
| | No | 81 | 21.50 |
| | Not sure | 134 | 35.50 |

**(The total participants were 377 unless otherwise stated for the purpose of the analysis; MOH Ministry of Health)**

**Table 5. Factors associated with COVID-19 vaccine acceptance among participants.**

| Variables | Category | COR (95% CI) | p-value | AOR (95% CI) | p value |
|---|---|---|---|---|---|
| **Age (years)** | < 25 | Ref | | Ref | |
| | ≥25 | 2.4(1.4–4.0) | 0.001* | 2.1(1.1–4.1) | 0.033* |
| **Sex** | Male | Ref | | Ref | |
| | Female | 1.4(0.8–2.3) | 0.209 | 2.0(1.1–3.6) | 0.019* |
| **Education** | Basic education | Ref | | Ref | |
| | SHS and above | 5.0(1.5–16.4) | 0.008* | 3.3(1.8–13.0) | 0.044* |
| **Occupation** | Unemployed | Ref | | Ref | |
| | Employed | 4.3(2.5–7.2) | <0.001* | 2.5(1.3–4.8) | 0.005* |
| **COVID-19 is real** | No | Ref | | Ref | |
| | Yes | 3.5(1.1–11.7) | 0.041* | 2.6(0.7–9.5) | 0.155 |
| **Heard of Immunity** | No | Ref | | Ref | |
| | Yes | 2.5(0.9–6.5) | 0.062 | 1.3(0.4–3.7) | 0.681 |
| **Vaccination gives immunity** | No | Ref | | Ref | |
| | Yes | 1.9(1.1–3.2) | 0.013* | 1.1(0.6–2.0) | 0.700 |
| **Vaccine prevents severity of disease** | No | Ref | | Ref | |
| | Yes | 1.6(0.9–2.7) | | 1.1(0.6–2.1) | 0.721 |
| **Been positive for COVID-19** | No | Ref | | Ref | |
| | Yes | 2.2(0.9–5.5) | 0.078 | 1.9(0.7–5.3) | 0.236 |
| **Heard of new vaccine** | No | Ref | | Ref | |
| | Yes | 4.4(2.3–8.5) | <0.001* | 2.9(1.4–6.0) | 0.004* |

Ref = Reference category;

*significance value; COR = Crude Odds Ratio; AOR = Adjusted Odds Ratio; Basic education is used to refer to six years of elementary education and three years of junior secondary/high education.

education and promotion efforts by the Ghana Health Service and its partners. Education and promotional activities should consider the varying educational levels of the population.

Knowledge of COVID-19 and its vaccines showed that most of the participants indicated that COVID-19 was real. However, we noticed that a significant proportion of vaccine hesitancy was among participants who indicated that COVID-19 was real. This may mean that just acknowledgement of the existence of the infection was not enough to influence individuals to accept the COVID-19 vaccine. Significant proportions of participants reported knowledge of immunity, vaccination giving immunity, vaccination not preventing the occurrence of other diseases, vaccine preventing the severity of disease, and awareness of new vaccines. In all these observations, vaccine hesitancy was common among persons who portrayed greater knowledge. This further suggests that knowledge of the COVID-19 and its vaccines alone may not be enough to influence people to vaccinate. Importantly, this observation could be largely

attributed to the high educational background of the participants. Our study findings on knowledge of COVID-19 and its vaccines were higher compared to another study conducted in Navrongo, Ghana [28]. In a study, which used a cross-sectional approach to examine COVID-19 vaccination intentions among literate Ghanaians, it was revealed that there is a low level of knowledge about the safety and efficacy of the vaccines. Only 45.2% of the participants indicated that the COVID-19 vaccine was effective [28]. Nonetheless, addressing the challenge of COVID-19 vaccine hesitancy to boost acceptance should go beyond creating awareness. It should involve to the institutionalization of policies that requires individuals to be vaccinated in order to achieve herd immunity.

The perception of participants toward COVID-19 vaccines could influence their uptake [28,29]. Our study showed that participants highly perceived they were at low risk of being severely infected with COVID-19. Most of the participants believed that they were not very susceptible to contracting COVID-19 and they thought that the disease could be severe if they did contract it. These findings could explain the low uptake of the COVID-19 vaccine among participants. According to Awuni et. al. [28], the majority of the participants believed that the COVID-19 pandemic could be eradicated through effective adherence to preventive measures without relying on vaccinations.

Some common reasons for participants refusing the COVID-19 vaccines include mistrust of the source of the vaccine, personal beliefs and experience, mistrust of the drug development process, mistrust of the health system, and mistrust of the pharmaceutical companies. The mistrust associated with vaccines has been a significant barrier to global vaccine acceptance and uptake [1,28,30,31]. The issue of COVID-19 vaccines leading to hesitancy needs to be addressed through transparent vaccine development and deployment. National food and drug authorities should play a vital supervisory and regulatory role to ensure that vaccines are developed with high safety standards. This should involve ensuring that information about current vaccines is easily accessible to the public. It is crucial to provide comprehensive information about the COVID-19 vaccines and share positive experiences from vaccinated individuals. Common sources of information cited by our participants, such as television, social media, radio, health workers, and newspapers can be utilised to disseminate important details about COVID-19 and the available vaccines to the public. The importance and use of this common information have been highlighted in other studies [31–33].

Though just a small proportion of participants portrayed misconception, this proportion is significant enough to influence others, especially those who are not sure and do not have adequate knowledge about COVID-19 and its vaccines. These misconceptions and misinformation about COVID-19 vaccines have been reported in other studies [28,29]. The misconceptions have to be corrected swiftly if the Central Region is to reduce the 49.6% of its target population who are yet to be vaccinated against the COVID-19 pandemic [13].

Participants indicated COVID-19 post-vaccination strategies should include the use of sanitizer, washing of hands, keeping social distance, and avoiding hand shake. These findings are particularly important to reduce the risk of COVID-19 reinfection after vaccination. The public should be engaged in the need to continue to adhere to the COVID-19 safety protocols as we are still not out of the woods. Though COVID-19 vaccine hesitancy was common among the participants, our study showed that most are willing to recommend the vaccine to others. This means that if we address key prevailing factors as the basis for refusal to vaccinate, the vaccinated individuals could influence a lot more to also vaccinate.

Among the predictors for COVID-19 vaccine acceptance include the unadjusted odds ratio analysis, which revealed that participants aged 25 years and older, those with educational level of Senior High School (SHS) and above, as well as those who are employed, believe that COVID-19 is real, understand that COVID-19 vaccination provides immunity, and have

heard of a new vaccine are more likely to accept the COVID-19 vaccine. The adjusted odds ratio confirmed that participants aged 25 years and older, female, educational level from SHS and above, being employed, and hearing of information about new vaccine influenced on COVID-19 vaccine acceptance. To effectively address the challenge of COVID-19 vaccine hesitancy, Ghana's Ministry of Health, and the Ghana Health Service should incorporate these factors identified to re-plan COVID-19 and its vaccines health campaigns in urban areas, taking into consideration the various age groups, educational background, gender and occupation and knowledge of vaccines.

## Strengths and Limitations

Several limitations are worth noting in interpreting the findings: The sample size calculation provided in the methods section was not adjusted for the multistage clustering. Consequently, it may be inadequate or underpowered. Additionally, the calculation did not account for the design effect, potentially leading to underestimation and insufficient statistical power. The intracluster correlation, which measures the similarity of responses within a cluster, was also not considered, possibly affecting the sample size.

The response rate was 93.1%, which is below the estimated sample. This may impact the generalizability of the study. The study collected data at one point in time, and the COVID-19 vaccine acceptance rate may have changed since the study was conducted. Again, the study was urban in nature and does not cover the rural towns of the Central Region of Ghana. The factors identified may be context-specific to urban area and may not reflect those in rural and smaller towns.

Despite these limitations, the study can still provide valuable insights for public health interventions in Cape Coast. A population-based study, it presents findings from diverse groups of individuals.

## Conclusion

The acceptance of COVID-19 vaccine was low, with high level of vaccine hesitancy among the participants. Common reasons for this hesitancy included mistrust of the vaccine source, personal beliefs and experiences, mistrust of the drug development process, mistrust in the health system, and mistrust of pharmaceutical companies. Individuals who were 25 years and older, female, had a high school education or above, were employed, and had heard about new vaccines were significantly more likely to accept the COVID-19 vaccine. The Ministry of Health in Ghana and its partners should reconsider their urban COVID-19 vaccine campaigns to address misconceptions and misinformation in order to increase vaccine acceptance.

## Supporting information

**S1 File. COVID-19 dataset.**
(XLSX)

## Acknowledgments

The authors wish to extend their appreciation to the study participants for making time off their busy schedule to be part of the study. We are grateful to Jeff Agyepong Adjei and colleagues for assisting with the data collection.

## Author contributions

**Conceptualization:** Hannah Benedicta Taylor-Abdulai, Stephen Ocansey, Samuel Kofi Arhin, Victor Obiri Opoku, Zakariah Jirimah Mankir, Sylvester Ackah Famieh.

**Data curation:** Hannah Benedicta Taylor-Abdulai, Victor Obiri Opoku, Zakariah Jirimah Mankir.

**Formal analysis:** Edem Kojo Dzantor, Mubarick Nungbaso Asumah.

**Investigation:** Hannah Benedicta Taylor-Abdulai, Stephen Ocansey, Samuel Kofi Arhin, Precious Barnes, Victor Obiri Opoku, Zakariah Jirimah Mankir, Sylvester Ackah Famieh, Collins Paa Kwesi Botchey.

**Methodology:** Hannah Benedicta Taylor-Abdulai, Nathan Kumasenu Mensah, Stephen Ocansey, Samuel Kofi Arhin, Precious Barnes, Victor Obiri Opoku, Zakariah Jirimah Mankir, Sylvester Ackah Famieh, Collins Paa Kwesi Botchey.

**Project administration:** Hannah Benedicta Taylor-Abdulai, Stephen Ocansey, Victor Obiri Opoku, Sylvester Ackah Famieh.

**Software:** Edem Kojo Dzantor, Mubarick Nungbaso Asumah.

**Supervision:** Hannah Benedicta Taylor-Abdulai, Nathan Kumasenu Mensah, Precious Barnes.

**Validation:** Hannah Benedicta Taylor-Abdulai, Nathan Kumasenu Mensah.

**Visualization:** Edem Kojo Dzantor, Nathan Kumasenu Mensah, Mubarick Nungbaso Asumah, Precious Barnes, Collins Paa Kwesi Botchey.

**Writing – original draft:** Hannah Benedicta Taylor-Abdulai, Edem Kojo Dzantor, Mubarick Nungbaso Asumah.

**Writing – review & editing:** Nathan Kumasenu Mensah, Stephen Ocansey, Samuel Kofi Arhin, Precious Barnes, Sylvester Ackah Famieh, Collins Paa Kwesi Botchey.

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
