## [Decision Letter · Decision Letter 0]

2 Jul 2023

PONE-D-23-13433Coronavirus disease 2019 (COVID-19) vaccine acceptability in Ghana: An urban-based population studyPLOS ONE

Dear Dr. Mensah,

Thank you for submitting your manuscript to PLOS ONE. After careful consideration, we feel that it has merit but does not fully meet PLOS ONE’s publication criteria as it currently stands. Therefore, we invite you to submit a revised version of the manuscript that addresses the points raised during the review process.

We look forward to receiving your revised manuscript.

Kind regards,

Md. Golam Dostogir Harun, MSS, MPH

Academic Editor

PLOS ONE

3. Please amend your manuscript to include your abstract after the title page.

Additional Editor Comments:

This manuscript required major revision and I encourage the authors to address point by point of reviewer comments and submit the updated version of the manuscript for further evaluation.

Reviewers' comments:

Reviewer's Responses to Questions

**Comments to the Author**

1. Is the manuscript technically sound, and do the data support the conclusions?

Reviewer #1: Partly

2. Has the statistical analysis been performed appropriately and rigorously? 

Reviewer #1: No

3. Have the authors made all data underlying the findings in their manuscript fully available?

Reviewer #1: No

4. Is the manuscript presented in an intelligible fashion and written in standard English?

Reviewer #1: No

5. Review Comments to the Author

Reviewer #1: PONE-D-23-13433

Coronavirus disease 2019 (COVID-19) vaccine acceptability in Ghana: An urban based population study

This manuscript needs significant improvement. The study rationale is poorly articulated and doesn’t answer the “So What” question.

Abstract:

The study site and the time of the study are missing.

How the study was implemented is missing. How the respondents received the questionnaires?

Results: The proportion could be presented and the numerator and denominator could be presented in parenthesis. The denominator should be mentioned here. Vaccine hesitancy 301 (80%) -what is the denominator here.

“Out of the 377 participants, their Socio-demographic characteristics showed that the majority were below 25 years (203, 53.8%) [vaccine acceptance; 28(13.8%) vs vaccine hesitancy; 175(86.2%)], and Females (189, 50.1%) [vaccine acceptance; 43(22.8%) vs vaccine hesitancy; 146(77.2%)].” The way results are presented is difficult to understand. Mistrust, Personal- the capital letter should be avoided.

Conclusion

“COVID-19 vaccine acceptance was low with high vaccine hesitancy among participants”. Its already known. What new knowledge this paper is going to add to the literature. Di you find any factors that mostly contributing to this low acceptance.

Manuscript

Methods

Data collection and procedures

“The questionnaires were interviewer assisted and respondents were assured that their participation was entirely voluntary”. In the abstract you have mentioned self-administered questionnaire.

Processing and statistical analysis

“Finally, Multivariate Logistic Regression Model was performed with selected variables that had significant relationship at the chi-square level with odds ratios (OR) and their 95% confidence intervals (95% 131 CI). A p-value of less than 0.05 was considered statistically significant”. What criteria was followed to select variables for the multivariate model?

“Had significant relationship at the chi-square level with odds ratios (OR)” not sure what do you mean by that. Do you mean the variable was significant in both chi-square and logistic regression analysis? It’s now clear how did you get this OR.

Results-The data should be presented in a clear way. <25 years…Its not clear who the comparison group here and why the authors divided them in this way. For religion and education, there were multiple categories and it’s not clear what is the comparison group when they presented Christians (334, 88.6%) [vaccine acceptance; 67(20.1%) vs vaccine hesitancy and attained tertiary education (253, 67.1%) [vaccine acceptance; 65(25.7%) vs vaccine hesitancy.

Table 4 should be formatted in a professional way.

Table 5: What is COR? Why did you exclude religion from the model. Why was education categorised into binary? What do you mean by basic education?

When you conduct a multivariate logistic regression, you need to present the adjusted odds ratio, not the crude ratio.

Discussion

Lines 254-255 “The purpose of this study was to assess COVID-19 vaccine acceptance among urban dwellers in 255 Cape Coast, Central Region, Ghana” repetition of information.

Lines 347-349, “Age ≥25 years, female, educational level SHS and above, being employed, and hearing of new vaccine had a significant influence on COVID-19 vaccine acceptance”- repetition of information.

6. PLOS authors have the option to publish the peer review history of their article (what does this mean? ). If published, this will include your full peer review and any attached files.

**Do you want your identity to be public for this peer review?** For information about this choice, including consent withdrawal, please see our Privacy Policy .

Reviewer #1: No

---

## [Author Response · Author response to Decision Letter 1]

12 Aug 2023

Response to editor and reviewer comments have been uploaded according

---

## [Decision Letter · Decision Letter 1]

2 Nov 2023

PONE-D-23-13433R1Coronavirus disease 2019 (COVID-19) vaccine acceptability in Ghana: An urban-based population studyPLOS ONE

Dear Dr. Mensah, 

Thank you for submitting your manuscript to PLOS ONE. After careful consideration, we feel that it has merit but does not fully meet PLOS ONE’s publication criteria as it currently stands. Therefore, we invite you to submit a revised version of the manuscript that addresses the points raised during the review process.

We look forward to receiving your revised manuscript.

Kind regards,

Stephen Dajaan Dubik, BSc, MPH, MPhil

Academic Editor

PLOS ONE

Journal Requirements:

Reviewers' comments:

Reviewer's Responses to Questions

**Comments to the Author**

1. If the authors have adequately addressed your comments raised in a previous round of review and you feel that this manuscript is now acceptable for publication, you may indicate that here to bypass the “Comments to the Author” section, enter your conflict of interest statement in the “Confidential to Editor” section, and submit your "Accept" recommendation.

Reviewer #2: All comments have been addressed

2. Is the manuscript technically sound, and do the data support the conclusions?

Reviewer #2: Yes

3. Has the statistical analysis been performed appropriately and rigorously? 

Reviewer #2: Yes

4. Have the authors made all data underlying the findings in their manuscript fully available?

Reviewer #2: Yes

5. Is the manuscript presented in an intelligible fashion and written in standard English?

Reviewer #2: Yes

6. Review Comments to the Author

Reviewer #2: The authors reported mutlistage cluster sampling however the sample size calculation provided in the methods section is not adjusted for the multistage clustering. Normally, the sample size is multiplied by the design effect to get the required power for the multistage cluster surveys. The given sample size is only based on simple random sampling and hence not adequate or underpowered. At this stage, the authors can only highlight this as an important limitation in the discussion section.

In addition, the authors have not adequately described the sample size calculation. Assumptions e.g., what proportion of COVID-19 vaccine hesitancy and acceptance was assumed to calculate the given sample size.

7. PLOS authors have the option to publish the peer review history of their article (what does this mean? ). If published, this will include your full peer review and any attached files.

**Do you want your identity to be public for this peer review?** For information about this choice, including consent withdrawal, please see our Privacy Policy .

Reviewer #2: **Yes: ** Mohammad Tahir Yousafzai

---

## [Author Response · Author response to Decision Letter 2]

9 Nov 2023

All comments have been addressed in the file response to reviewers comment

---

## [Decision Letter · Decision Letter 2]

1 Jul 2024

PONE-D-23-13433R2Coronavirus disease 2019 (COVID-19) vaccine acceptability in Ghana: An urban-based population studyPLOS ONE

Dear Dr. Mensah,

Thank you for submitting your manuscript to PLOS ONE. After careful consideration, we feel that it has merit but does not fully meet PLOS ONE’s publication criteria as it currently stands. Therefore, we invite you to submit a revised version of the manuscript that addresses the points raised 

We look forward to receiving your revised manuscript.

Kind regards,

Khin Thet Wai, MBBS, MPH, MA

Academic Editor

PLOS ONE

Additional Editor Comments:

To revise in line with reviewers' comments particularly the Methodological section. Some of the issues pointed out need to be addressed as limitations.

Reviewers' comments:

Reviewer's Responses to Questions

**Comments to the Author**

1. If the authors have adequately addressed your comments raised in a previous round of review and you feel that this manuscript is now acceptable for publication, you may indicate that here to bypass the “Comments to the Author” section, enter your conflict of interest statement in the “Confidential to Editor” section, and submit your "Accept" recommendation.

Reviewer #3: (No Response)

Reviewer #4: (No Response)

2. Is the manuscript technically sound, and do the data support the conclusions?

Reviewer #3: Partly

Reviewer #4: Partly

3. Has the statistical analysis been performed appropriately and rigorously? 

Reviewer #3: I Don't Know

Reviewer #4: Yes

4. Have the authors made all data underlying the findings in their manuscript fully available?

Reviewer #3: No

Reviewer #4: Yes

5. Is the manuscript presented in an intelligible fashion and written in standard English?

Reviewer #3: No

Reviewer #4: Yes

6. Review Comments to the Author

Reviewer #3: This is the first time that I am seeing this manuscript.

GENERAL FEEDBACK

Inclusion or reference to a specific Theoretical Framework that assesses vaccine acceptability would be useful in grounding your findings.

SECTION FEEDBACK:

DATA QUALITY CONTROL - This section needs development to enhance clarity and transparency.

Data quality control - "Data were collected under strict supervision"

Please provide detail on the data collection processes, e.g. number of data collectors, the paper data collection method used, data collector training, contextual piloting of the tool, spot checks, data checks, data cleaning practices, etc.

DATA PROCESSING AND STATISTICAL ANALYSIS

..."Data were analyzed using appropriate descriptive and inferential statistics." Please expand on this statement so that readers have confidence that "appropriate" analysis practices were followed.

DISCUSSION

New findings are introduced in the discussion section. The discussion section should generally include discussion of findings presented in the previous FINDINGS or RESULTS section.

Also a number of studies are cited, however, no detail on study design, context or findings for the cited studies is included. Please provide more cited study background in order to contextualise and support the relevance of the cited studies for discussion and comparison to this acceptability study.

Reviewer #4: The study aims to assess COVID-19 vaccine acceptance among urban dwellers in Cape Coast, Central Region, Ghana. This is contextualized within the broader public health challenge of vaccine hesitancy and the specific socio-demographic factors influencing vaccine uptake in this region.

Few notable limitations undermining the study relevance are :

1. The sample size calculation provided in the methods section is not adjusted for the multistage clustering and hence may be inadequate or underpowered.

2. Proportion Estimates: Assumptions about the proportion of vaccine acceptance and hesitancy should have been clearly stated and justified. The sample size calculation should reflect these assumptions and be adjusted for clustering effects.

3. Finite Population Correction: When sampling from a finite population, a finite population correction factor can be applied to adjust the sample size. The study did not address whether this was considered.

4. Stratification: The study mentioned stratification of zones but lacked detail on how this stratification was done and how it ensured representativeness.

5. Selection Process: Detailed description of the selection process at each stage is crucial. The study should have provided more information on how households and individuals within households were selected to avoid selection bias.

6. Design Effect (DEFF): This is a factor that accounts for the increased variance that typically occurs in cluster sampling compared to simple random sampling. The study did not adjust the sample size calculation for the design effect, which can lead to underestimation of the required sample size and, consequently, insufficient statistical power.

7. Intracluster Correlation: The ICC measures the similarity of responses within clusters. Higher ICC values indicate that individuals within clusters are more similar to each other than to those in other clusters. The study did not mention any assumptions or considerations for ICC, which is essential for calculating the design effect and adjusting the sample size accordingly.

7. PLOS authors have the option to publish the peer review history of their article (what does this mean? ). If published, this will include your full peer review and any attached files.

**Do you want your identity to be public for this peer review?** For information about this choice, including consent withdrawal, please see our Privacy Policy .

Reviewer #3: **Yes: ** Susan A. Kelly, PhD MPH

Reviewer #4: **Yes: ** Neeti Rustagi

---

## [Author Response · Author response to Decision Letter 3]

30 Jul 2024

Dear Editor,

I have addressed all the comments and have attached the requires files

---

## [Decision Letter · Decision Letter 3]

10 Feb 2025

Coronavirus disease 2019 (COVID-19) vaccine acceptability in Ghana: An urban-based population study

PONE-D-23-13433R3

Dear Dr. Mensah,

We’re pleased to inform you that your manuscript has been judged scientifically suitable for publication and will be formally accepted for publication once it meets all outstanding technical requirements.

Kind regards,

Mona Gamal Mohamed

Academic Editor

PLOS ONE

Additional Editor Comments (optional):

Reviewers' comments:

Reviewer's Responses to Questions

**Comments to the Author**

1. If the authors have adequately addressed your comments raised in a previous round of review and you feel that this manuscript is now acceptable for publication, you may indicate that here to bypass the “Comments to the Author” section, enter your conflict of interest statement in the “Confidential to Editor” section, and submit your "Accept" recommendation.

Reviewer #3: All comments have been addressed

Reviewer #5: All comments have been addressed

2. Is the manuscript technically sound, and do the data support the conclusions?

Reviewer #3: Yes

Reviewer #5: Yes

3. Has the statistical analysis been performed appropriately and rigorously? 

Reviewer #3: I Don't Know

Reviewer #5: Yes

4. Have the authors made all data underlying the findings in their manuscript fully available?

Reviewer #3: Yes

Reviewer #5: Yes

5. Is the manuscript presented in an intelligible fashion and written in standard English?

Reviewer #3: Yes

Reviewer #5: Yes

6. Review Comments to the Author

Reviewer #3: The authors have addressed the comments I provided on the previous version of this article. I suggest that another reviewer comment on the soundness of the statistical methodology as I am a qualitative research working in the field of vaccine acceptability, i.e. not an epidemiologist/statistician.

I would suggest that the authors have the manuscript read through by someone for grammar and proofreading before final submission as I saw a few editorial issues. Also data are typically referred to as plural, i.e. "data are" vs. "data is" so looking through for grammatical consistency when discussing data would also be useful.

Reviewer #5: Attached word, file. may be seen for better understanding of following comments

Line#89 Name of the author or study may be added, Add Text, to increase clarity

Line#231 Vaccine acceptance was significant among those with knowledge of vaccination giving immunity, Add Text, To increase clarity

Line#244 Social Media, since all words start from uppercase, Add Text, for Formatting

Line#389 level from SHS and above, being employed, and information about new vaccines influenced on COVID-19 vaccine acceptance., Change Text, to increase clarity

Line#405 Despite these limitations, the study can still provide valuable insights for public health interventions in Cape, Add Text,

7. PLOS authors have the option to publish the peer review history of their article (what does this mean? ). If published, this will include your full peer review and any attached files.

**Do you want your identity to be public for this peer review?** For information about this choice, including consent withdrawal, please see our Privacy Policy .

Reviewer #3: **Yes: ** Susan A. Kelly, PhD MPH

Reviewer #5: **Yes: ** Dr Mahvash Ansari

---

## [Editor Report · Acceptance letter]

PONE-D-23-13433R3

PLOS ONE

Dear Dr. Mensah,

I'm pleased to inform you that your manuscript has been deemed suitable for publication in PLOS ONE. Congratulations! Your manuscript is now being handed over to our production team.

Kind regards,

on behalf of

Dr. Mona Gamal Mohamed

Academic Editor

PLOS ONE